# A Study on the Impact of External Shocks on the Resilience of China's Grain Supply Chain

**Tao Zheng [1,2], Guiqian Zhao [1,*] and Siwei Chu [1]**

1   School of Economics and Management, Yanshan University, Qinhuangdao 066004, China;
    ztao@ysu.edu.cn (T.Z.); csw751143874@163.com (S.C.)
2   The Regional Economic Development Research Center, Yanshan University, Qinhuangdao 066004, China
*   Correspondence: zgqian123@163.com

**Abstract:** Grain supply is the lifeblood of a country, and the stability of the supply chain is a crucial prerequisite for ensuring national grain security. This paper draws on the definition of resilience in physics and economics. It takes supply chain fracture resilience, impact resilience, and synergy resilience as the secondary indicators. It constructs a comprehensive evaluation indicator system of the grain supply chain resilience, measures the resilience indicator of China's grain supply chain from 1996 to 2021, and analyzes the role of supply, cost, exchange rate, and other external shocks in influencing the resilience of China's grain supply chain on this basis. The study found that the overall level of China's total grain supply chain resilience has been growing year by year and can be divided into three stages: low-level stabilization stage, continuous growth stage, and high-level stabilization stage. Grain supply chain fracture resilience has been growing steadily, grain supply chain impact resilience fluctuation is more obvious, and grain supply chain synergy resilience has been changing more gently. In the inquiry of the impact of external shocks on the resilience of China's grain supply chain, it was found that world grain exports and the RMB exchange rate have a significant positive impact on China's grain supply chain resilience level, while the international oil price has a significant negative impact. Based on this, the paper puts forward suggestions for ensuring stable production and supply in the grain market, improving the structure of foreign trade in grain, and actively coping with international commodity price shocks.

**Keywords:** resilience; grain supply chain; external supply shock; external cost shock; exchange rate volatility shock

## 1. Introduction

Grain is the material basis for the sustainable development of the national economy and society, providing the prerequisites for national stability and the active and healthy production and living of the population [1]. As the world's most populous developing country, with agriculture as its key strategic base, China has always attached great importance to grain security. Grain security is not only a matter of national economic development and social stability, but also a matter of national security and national self-reliance. Since 2013, China's grain production has been maintaining stable development, with output steadily increasing and the linkages between the various links in the supply chain gradually being strengthened. However, it should also be realized that China's grain supply chain is facing challenges due to a large population base, limited arable land and water resources, especially the frequent occurrence of natural disasters, and other factors. In early 2020, the novel coronavirus (COVID-19) epidemic caused significant fluctuations in global grain market prices, posing a major challenge to China's grain circulation [2]. In early 2022, the Russia—Ukraine war broke out, intensifying international agricultural trade patterns and grain price shocks. This further tightened the global grain supply chain, leading to a gradual increase in China's grain, fertilizer, and feed prices. Against this backdrop, China's

grain supply chain has witnessed unconventional incidents of poor operating systems and even faces a crisis of chain breakage, seriously jeopardizing grain security. Strengthening the resilience of the grain supply chain and promoting the stability and safe operation of the grain supply chain have gradually become hotspots of concern for all walks of life.

The current world pattern has undergone profound changes, and the international environment is unpredictable [3]. China's grain industry should not only adapt to the new normal of continuous economic decline, but also need to pay attention to the impact that various types of black swan events may bring. In the face of major changes not seen in a hundred years, profound changes in the external environment, difficulties, and risks have increased significantly. To enhance the resilience of the grain supply chain can lay a solid foundation for the rapid recovery and high-quality development of China's agricultural economy. Grain supply chain resilience refers to the capacity of the grain supply chain to sustain stability and to adapt, recover, and transform following shocks. This resilience relies on inherent attributes, external safeguards, and internal linkages to rebound from external disruptions. According to the classification of supply chain resilience, this paper categorizes grain supply chain resilience into three parts: grain supply chain fracture resilience, grain supply chain imp resilience, and grain supply chain synergy resilience.

This paper takes the resilience of the grain supply chain as the research object, makes a comprehensive evaluation of the resilience of the grain supply chain, gives the time evolution characteristics of the resilience of China's grain supply chain, analyzes the impacts of the main external shock factors on the resilience of China's grain supply chain, and puts forward suggestions and strategies for enhancing the resilience of China's grain supply chain on this basis, in order to provide theoretical references for strengthening and stabilizing the chain of China's grain supply.

## 2. Literature Review

Based on concepts from physics and engineering, resilience initially referred to the ability of a system to return to its original state after being stressed [4]. In the perspective of economics, some scholars, based on the equilibrium theory, focus on how an economy can return to the initial equilibrium state after suffering a shock [5]. Currently, most scholars have studied economic resilience based on evolutionary theory, which views resilience as the ability of a region to achieve its sustainable development through the adjustment of its society, economy, and system in the face of environmental changes. The theory of evolutionary resilience suggests that there is not a stable equilibrium for regions, and Keith proposes that regional economic resilience is a transformational and upgrading capacity that enables changes in its own institutional and organizational structure when it is subjected to external shocks [6–10]. Currently, research on the grain supply chain mainly focuses on the aspects of risk assessment and control. Larson started from the cross-cutting and complexity of grain and put forward the risk of grain quality and safety in the whole process from production to consumption, pointing out the high risk of the grain supply chain [11]. Tah and others proposed an assessment method for the hierarchical risk of the grain supply chain in order to fill the deficiencies in the existing risk management process, management tools, and technological means [12]. Garcia and others focused on the nodes of the agricultural supply chain as well as the whole framework and constructed a structural model for the grain supply chain system, which is useful for realizing the risk management of it. The grain supply chain and the real-time traceability have a certain reference role [13]. Mu et al. propose assessing the resilience of food supply chains in terms of three dimensions: time, the degree of impact of food safety shocks, and the degree of recovery [14]. Panicker believes that grain is an essential necessity for human beings, which highlights the importance of risk control of the grain supply chain [15].

The literature on the resilience of China's grain supply chain is extremely scarce, and the relevant literature mainly focuses on the risk composition system, risk indicator weights, and risk dimensions of the grain supply chain. Chen reorganized the concept of the grain supply chain and its internal dimensions, summarized the risk characteristics

of the grain supply chain, and based on these characteristics, put forward the scientific proposal of constructing the emergency response linkage mechanism and risk early warning mechanism of the grain supply chain [16]. Ding and Yang summarized the characteristics of China's grain supply chain, identified eight potential risk factors, and provided relevant preventive measures [17]. Wang and others constructed a comprehensive indicator system of grain supply chain risks from different perspectives, scientifically evaluated the risk level of the grain supply chain, and arranged the root causes of various risks according to priorities [18].

Regarding resilience measurement, two methods are mostly used. One is the comprehensive indicator system method, which involves selecting specific indicators reflecting resilience through theoretical analysis to constitute the indicator evaluation system. Zeng attempted to establish a corresponding comprehensive evaluation indicator system from seven dimensions, including robustness, risk vulnerability, intrinsic stability, high liquidity, structural balance, innovativeness, and changeability [19]. Chen and Ding chose five indicators, namely, the degree of industrial agglomeration, the level of economic growth, the gap between the rich and the poor, the degree of optimization of the industrial structure, and the sensitivity of the urban economy, to examine the level of the city's economic resilience [20]. Liu and others established the resilience indicator of cities in the city cluster in the middle reaches of the Yangtze River. They established the indicator of urban resilience from four perspectives: economy, society, ecology, and urban construction. They used the entropy value method to measure and analyze the economic resilience of each prefecture in the city cluster in the middle reaches of the Yangtze River, resulting in the indicator of economic resilience of each prefecture-level city [21]. The other is the single indicator method, which selects a central variable that responds to external perturbations, and based on this, analyzes the gap value of this central variable under the influence of external perturbations. Most scholars use indicators such as GDP, employment and unemployment [22–24]. This article employs a comprehensive indicator method to assess the resilience of the grain supply chain. The comprehensive indicator system is a complete and scientifically sound evaluation system. At the same time, the resilience of the grain supply chain is complex and influenced by multiple factors. Therefore, employing a comprehensive indicator system to assess the resilience level of the grain supply chain is the most scientific and rigorous approach.

With the deepening of the global integration process, the synergistic effect of countries' economic development is gradually strengthened, and external shocks have become an important aspect of a country's macroeconomic development. Swanepoel selected crude oil prices, exchange rates, and the price of non-oil imports as the most important external shock variables to study their role in South Africa's economic development and came up with the relative importance of various external shocks [25]. Gosse and Guillaumin analyzed the shocks to the economies of Southeast Asia from three perspectives: the oil crisis, the currency crisis in the United States, and the fiscal crisis in the U.S. [26]. Anetor empirically analyzed the impact of foreign capital entry into Nigeria using quarterly information for the period 1986–2016 and gave the corresponding policy responses [27]. In the empirical analysis conducted by Jones, it was found that African countries are affected by crude oil shocks and increasing financial market volatility through trade and financial agglomeration. Among these modes of transmission, trade agglomeration was identified as the most significant in transmitting these exogenous shocks [28]. Olamide et al. argue that oil price shocks have a significant and negative impact on commodity output and growth [29]. Liu categorizes external shocks into two types based on their effects: inward and outward. Inward shocks impact various elements within the economic system, while outward shocks have a certain degree of influence on the transmission and penetration of the entire economic system [30]. According to Liu, in an open macroeconomic environment, the negative effects encountered by an economy caused by other economies, such as import and export trade deficits, rising crude oil prices, interest rate fluctuations, etc., are all exogenous shocks [31]. As for the effect of external shocks on a certain industry or

product, the existing literature mainly focuses on the price of agricultural products and grain price [32–34].

From the previous review, the current scholars on resilience research are mostly focused on regional economic resilience, urban resilience, and enterprise resilience perspective [7–10]. There are more studies from a macro or micro perspective, and fewer research results from the meso-industry perspective, especially regarding the resilience of the industrial supply chain. In terms of the research on the impact of external shocks, most of the current research on these issues is centered on the impact of external shocks on domestic prices, as well as the synergistic effect of external shocks and inter-regional or world economic fluctuations on this issue. There are fewer studies specifically studying the impact of external shocks on the resilience of China's grain supply chain. Therefore, this paper starts from a meso perspective, takes the grain supply chain as the research object, evaluates the resilience of the grain supply chain, and researches the degree of its influence by external shocks.

## 3. Theoretical Analysis and Hypotheses

*3.1. Grain Supply Chain Resilience*

As a complex industrial system, the grain supply chain, after being hit by external shocks, relies on the supply chain's own historical accumulation and resource endowment, as well as the supply chain's external safeguards and internal links, to form the ability to withstand shocks, rapidly recover, and transform and upgrade. Therefore, on the basis of the concept of supply chain resilience, this paper considers that the resilience of the grain supply chain is the ability of the grain supply chain to maintain a stable state and recover, adjust, and transform from shocks by relying on its inherent properties, external safeguards, and internal links after suffering an external shock.

In physics, fracture resilience is considered as a measure of a substance's ability to prevent the destabilizing expansion of macroscopic cracks, as well as a resilience parameter of a material to resist brittle damage. It does not have any relationship with the size and shape of the crack itself and the magnitude of the applied force. Fracture resilience is an inherent property of a material and is only related to the material itself in terms of its heat treatment and machining process [35]. Impact resilience is regarded as the ability of a material to absorb plastic deformation work and fracture work under impact loading, and it is a response to external impact loading [36].

According to the classification of supply chain resilience, the resilience of the grain supply chain is divided into three parts: the fracture resilience of the grain supply chain, the impact resilience of the grain supply chain, and the collaborative resilience of the grain supply chain. The fracture resilience of the grain supply chain is its intrinsic inherent property, which includes the volume scale, condition endowment, stability, and complexity within the system accumulated over a long period of time, just like the material organization in engineering, which belongs to its own intrinsic properties. The impact resilience of the grain supply chain is a deeper extension of the basic concept of physical impact resilience, which not only contains the negative impact but also contains the positive impact. The supply chain will also be affected by external security support and scientific and technological innovation support to enhance its disaster-resistant capacity. Comprehensively speaking, it is the impact of the system outside the system. The collaborative resilience of the grain supply chain is the expansion of the theory of synergy in the field of resilience, which includes the multidimensional impacts of grain production, acquisition, warehousing, processing, and sales. It includes multiple supply chain links, such as grain production, acquisition, storage, processing, and sales. The ability of resource integration and coordination between and within the links of the supply chain is reflected in the grain supply chain's ability to cope with internal and external changes.

### 3.2. Mechanisms of External Shocks on the Resilience of China's Grain Supply Chain

There are two main meanings of external shocks: the first is defined by the Dictionary of Modern Economics, compiled by Liu Shucheng of the Institute of Economics of the Chinese Academy of Social Sciences in 2005 [31], as the negative impact of a macroeconomy on the global economy in an open environment, and in particular, the large negative impact on the terms of trade of a given economy. Negative effects include, among others, a significant drop in demand for exports, a sharp rise in international interest rates, and a sharp increase in the prices of imported goods (especially those with a strong dependence on imports, such as crude oil). The second is the meaning of Modern Macroeconomic Shock Theory published by Jilin University Press in September 2000 [30], which divides economic fluctuations into two categories: those that arise from within the economic system and those that come from outside the economic system. An internal shock refers to an economic shock that occurs within an economic system (e.g., energy crisis, exchange rate changes, etc.), and most of its results are directed at factors internal to an economy, but it may also have an impact on the external aspects of an economy (e.g., social order), with some spillover effects. External economic shocks are those that occur outside the economic system (e.g., natural disasters, terrorist attacks, etc.) and they also affect factors within the economic system.

Combining the above two definitions and the research purpose of this paper, the external shocks in this paper include external supply shocks, cost change shocks, and exchange rate fluctuation shocks, among others. These shocks are not within the control of the grain supply chain system itself and have a significant impact on the overall supply system. This paper analyzes the impact mechanism of external shocks on the resilience of the grain supply chain through the above three levels of research, as shown in Figure 1.

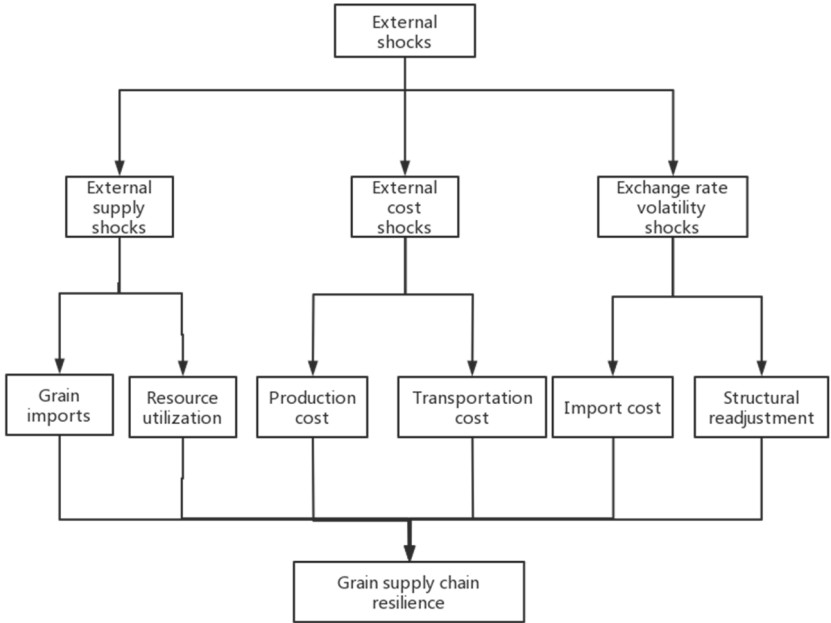

**Figure 1.** Mechanisms of external shocks on the resilience of China's grain supply chain.

The impact on our grain supply chain, as a result of external supply shocks, comes mainly from the international trade route [37]. At present, our country's external dependence on some grain varieties remains high, while the sources of grain imports are relatively concentrated. On the one hand, an increase in world grain exports can increase the sources of China's grain imports, thus reducing China's single dependence of on grain imports and guaranteeing the stability of China's grain supply. On the other hand, the volume of world grain exports allows China to use its limited land and water resources for the cultivation of other agricultural products, such as high-value-added agricultural products or grain-

processing raw materials, according to its resource advantages and climatic conditions. This can improve the efficiency of agriculture and the income of farmers. Accordingly, this paper proposes Hypothesis 1.

**H1.** *External supply shocks have a positive effect on grain supply chain resilience.*

Cost change shocks mainly refer to the shocks brought about by international commodity price fluctuations, and the transmission mechanism affecting the resilience of China's grain supply chain is mainly cost transmission. On the one hand, international crude oil prices affect the supply chain's generating capacity and, therefore, its resilience, by increasing production costs. As China's agricultural mechanization continues to advance, but when international crude oil prices continue to rise, the cost of grain production will continue to increase, thus affecting the level of supply chain resilience. On the other hand, by increasing the transportation cost, it affects the supply chain logistics capacity and thus affects the supply chain resilience. Since 2009, in the case of sustained high international oil prices, domestic crude oil prices have increased several times, resulting in an average annual increase of about 20% in the cost of grain transportation. In 2011, two additional crude oil price increases were experienced, which made the increase in the cost of oil transportation much greater than the cost before the oil price increase. Accordingly, this paper develops Hypothesis 2.

**H2.** *Cost variation shocks have a negative effect on grain supply chain resilience.*

The impact of exchange rate fluctuations is mainly on foreign liquidity, both from the perspective of international trade and financial markets, and will have an impact on the domestic grain supply chain. Because the U.S. dollar is the most important payment currency internationally, the RMB exchange rate mentioned in this paper refers to the ratio of the RMB to the U.S. dollar and is expressed using the indirect markup method. Changes in the RMB exchange rate have a certain impact on the resilience of China's grain supply chain. On the one hand, an appreciating exchange rate reduces the cost of imported grain, especially for agricultural products on which China relies for imports. It may provide more choices and a stable supply for regions that rely on imported grain. More imported grain can be purchased at a relatively lower cost, thus increasing domestic supply and easing the pressure on grain supply. On the other hand, it can promote agricultural restructuring. The appreciation of the RMB reduces the cost of imported grain and puts pressure on the competitiveness of domestic agricultural products, which helps to promote China's agricultural restructuring. The optimization and modernization of the agricultural structure will further enhance the resilience and self-sufficiency of China's grain supply chain and reduce its dependence on the international market. Accordingly, this paper proposes Hypothesis 3.

**H3.** *Exchange rate volatility shocks have a positive effect on grain supply chain resilience.*

## 4. Materials and Methods

### 4.1. The Construction of China's Grain Supply Chain Resilience Indicator System

Considering the use of the multivariate comprehensive indicator method is more comprehensive, this paper refers to the research of Cao De et al. and combines the origin of resilience, the connotation of resilience in the grain supply chain, and the theory of supply chain synergy when constructing the comprehensive indicator system for grain supply chain resilience [38]. The total resilience of the grain supply chain is taken as the primary indicator, and the supply chain fracture resilience, supply chain impact resilience, and supply chain synergy resilience are taken as three secondary indicators. The specific indicators are shown in Table 1.

**Table 1.** Indicator system for evaluating the resilience of grain supply chains.

| Primary Indicators | Secondary Indicators | Tertiary Indicators | Interpretation of Indicators | Unit | Indicator Attributes |
|---|---|---|---|---|---|
| Supply chains total resilience | Supply chain fracture resilience | Production capacity | Grain total output/average number of people in the primary industry | T/person | + |
| | | Warehousing capacity | Grain total output + net grain imports − grain sales volume | WT | + |
| | | Transport capacity | Total agricultural logistics | Trillion RMB | + |
| | | Processing capability | Finished goods/average number of workers | 10,000 RMB/person | + |
| | | Sales ability | Expressed in terms of the number of grain product movements, calculated as: grain sales/average grain stocks | % | + |
| | Supply chain impact resilience | Financial support | Financial support for agriculture | Billion RMB | + |
| | | Financial support | Balance of loans to financial institutions for manufacturing agricultural production materials in local and foreign currencies as a percentage of all loans | % | + |
| | | External Dependence | Grain imports/total grain production | % | - |
| | | R&D capability | Number of valid invention patents | PCS | + |
| | | Innovation capacity | Senior Agricultural Technician | Person | + |
| | | Capacity to translate results into action | Contractual turnover of the State Fund for the Transformation of Agricultural Science and Technology Achievements | Billion RMB | + |
| | | Disaster resistance | The area of the disaster/become the area of the disaster | % | + |
| | Supply chain synergy resilience | Synergy between production and marketing | Grain production in the year/grain sales in the year | % | + |
| | | Synergy between purchase and sale | Grain purchases in the year/grain sales in the year | % | + |
| | | Information synergy | Informatization development indicator | % | + |
| | | Transportation network | Road density | 1000 km | + |
| | | Circulation efficiency | Grain cargo turnover | WT | + |

Based on the availability of data, this paper selects the time series data related to the grain supply chain from 1996 to 2020 as a sample. The research data in this paper mainly come from the *China Statistical Yearbook*, *China Grain Yearbook*, *China Grain Network Database*, and *China Rural Statistical Yearbook*. Since the *China Grain Yearbook* was able to obtain the volume of grain purchases and grain sales from 1996–2016, but the statistical items of the subsequent data statistical yearbooks were changed, it was supplemented with the *China Grain and Material Reserve Yearbook*. In addition, the data on total agricultural logistics came from the *China Logistics Yearbook*; the data on financial support indicators came from the *China Financial Yearbook*; the data on R&D capacity indicators came from the *China Science and Technology Yearbook*; the data on innovation capacity indicators came from the *China Ethnic Statistical Yearbook*; the data on achievement transformation capacity indicators came from the *China Torch Statistical Yearbook*; and the information development indicator came from the *China Information Yearbook*. Due to the large number of indicators and the long timespan of the data required for the study, there is a small amount of missing data, which is supplemented by the linear interpolation method and regression fitting method.

### 4.2. Model Construction

To further examine the external shock factors on the resilience of China's grain supply chain, the following econometric model is constructed.

$$res_t = \alpha_0 + \alpha_1 exp_t + \alpha_2 oil_t + \alpha_3 ex_t + \alpha_4 X_t + \varepsilon_t \tag{1}$$

In model (1), the subscript *t* represents the year. *res* represents the total resilience indicator of the grain supply chain. *exp* represents the total world grain export. *oil* represents the international oil price. *ex* represents the RMB exchange rate. *X* represents the control variable. $\alpha_0$ is the error item. $\varepsilon$ is the intercept term. $\alpha_1$, $\alpha_2$, $\alpha_3$, $\alpha_4$ represent the explanatory variable coefficients.

*4.3. Selection of Variables*

- The dependent variable is grain supply chain resilience, which is represented in this paper using the measured total grain supply chain resilience indicator.
- The core explanatory variables are the volume of world grain exports, international oil prices, and the RMB exchange rate. World grain export volume (*exp*) is an external supply shock factor. Generally speaking, under open economy conditions, the supply factors that affect the grain supply chain depend on China's external grain dependence. Therefore, referring to the research of Wang Baihao, this paper uses the world grain export volume as the external supply shock [39]. International oil price (*oil*) is the cost change shock element. Production cost is usually considered as the main supply factor leading to changes in the resilience of China's grain supply chain. On the one hand, oil price affects supply chain resilience by pushing up production costs. As China's agricultural mechanization continues to advance, but when the price of oil continues to rise, the cost of grain production will continue to increase, thus affecting the level of supply chain resilience. On the other hand, it affects supply chain resilience by increasing transportation costs. Therefore, referring to Zhang Liang's study [40], this paper uses the international oil price as an external cost shock. The RMB exchange rate (*ex*) reflects the exchange rate fluctuation shock, and this paper selects the United States as the representative of the external economy to measure the RMB exchange rate indicator. This paper adopts the indirect valuation method, i.e., the exchange rate is expressed by the quantity of 1 unit of RMB against the US dollar (the following RMB exchange rates are all indirectly valued). An increase in the value of the exchange rate indicates RMB appreciation, while a decrease in the value indicates RMB depreciation, which is conducive to simplifying the understanding of the empirical results. Therefore, referring to Guo Fengjuan, this study utilizes the RMB exchange rate to reflect exchange rate volatility shocks [41].
- Control variables include urbanization rate (*urban*), industrial structure rationalization (*ind*), investment in rural fixed assets (*inv*), total reservoir capacity (*cap*), and gas price (*gas*). According to Hao Aimin, using the ratio of the urban population to the total population to represent the urbanization rate (*urban*) [42]. The rationalization of industrial structure is used to measure the rational distribution and efficient use of resource factors, with the formula $TL = \sum_{i=1}^{n} (Y_i/Y) Ln[(Y_i/L_i)/(Y/L)]$. The *TL* indicates the Taylor indicator, where the smaller the value indicates a more reasonable structure. $Y_i$ and $Y$ indicate the output value and output value of each industry, $L_i$ and $L$ indicate the labor input and total labor force of each sub-industry. The $n$ indicates the number of industry sectors. In this article, $n = 3$ represents the three-time industry. Reservoirs play an important role in agricultural irrigation, providing farmers with benefits in terms of drought mitigation and grain security. Therefore, the total reservoir capacity is chosen in this paper to measure the impact on the resilience of the grain supply chain. Natural gas plays an important role in grain production, processing, and transportation. Therefore, this paper chooses the natural gas price to measure the impact on the resilience of the grain supply chain.

*4.4. Data Sources*

The data used in this chapter are annual data from 1996 to 2021, including the world grain export volume (*exp*) from the website of the Food and Agriculture Organization of the United Nations, the urbanization rate (*urban*) from the *China Statistical Yearbook*, investment in rural fixed assets (*inv*), and the total reservoir capacity (*cap*) from the *China Agricultural Statistical Yearbook*, the international oil price (*oil*) and gas price (*gas*) from the Crude Oil and Petroleum Prices Network, and the RMB exchange rate (*ex*) from the World Bank database. There are numerous variables involved in the regression analysis, and some of them do not have quarterly data available. Additionally, China's reform and opening up occurred relatively late, and there is missing data for the period before 1996. Therefore, this article utilizes time series data from 1996 to 2021. In order to eliminate the heteroskedasticity

present in the time series and to linearize the trend, some of the variables are therefore subjected to a natural logarithmic transformation. The descriptive statistics of the main variables are shown in Table 2.

**Table 2.** Descriptive statistics of main variables.

| Variable | Connotation | Mean | Me | Max | Min | SD |
|---|---|---|---|---|---|---|
| res | Total grain supply chain resilience | 0.3447 | 0.34 | 0.65 | 0.12 | 0.186 |
| lnexp | World grain exports | 1.3806 | 1.33 | 1.86 | 0.88 | 0.308 |
| lnoil | International oil price | 3.8611 | 3.98 | 4.60 | 2.67 | 0.568 |
| ex | RMB exchange rate | 0.1383 | 0.14 | 0.16 | 0.12 | 0.016 |
| urban | Urbanization rate | 0.4756 | 0.48 | 0.65 | 0.30 | 0.102 |
| ind | Rationalization of industrial structure | 5.6329 | 4.51 | 11.04 | 3.10 | 2.377 |
| lninv | Investment in rural fixed assets | 7.7114 | 7.92 | 10.02 | 4.69 | 1.776 |
| lncap | Total reservoir capacity | 8.8180 | 8.85 | 9.20 | 8.43 | 0.249 |
| lngas | Price of natural gas | 3.2787 | 3.24 | 4.27 | 2.64 | 0.480 |

## 5. Results

### 5.1. Results and Trend Analysis of China's Grain Supply Chain Resilience Measurements

This study adopts the entropy weight method to measure the evaluation indicators of China's grain supply chain resilience. The entropy weight method can measure the indicator weights according to the real value of each indicator data in the sample, thus effectively avoiding the weight bias caused by subjective bias, and is suitable for the comprehensive evaluation of the multivariate indicator system, including the specific steps of data standardization, measuring entropy value, calculating the coefficient of variation, and calculating the weights. Specifically, according to the grain supply chain toughness evaluation index system constructed in Table 1 above, the relevant calculation formula is applied to weigh the Chinese grain supply chain toughness index. The calculated growth trend of China's grain supply chain resilience index is shown in Figure 2 below.

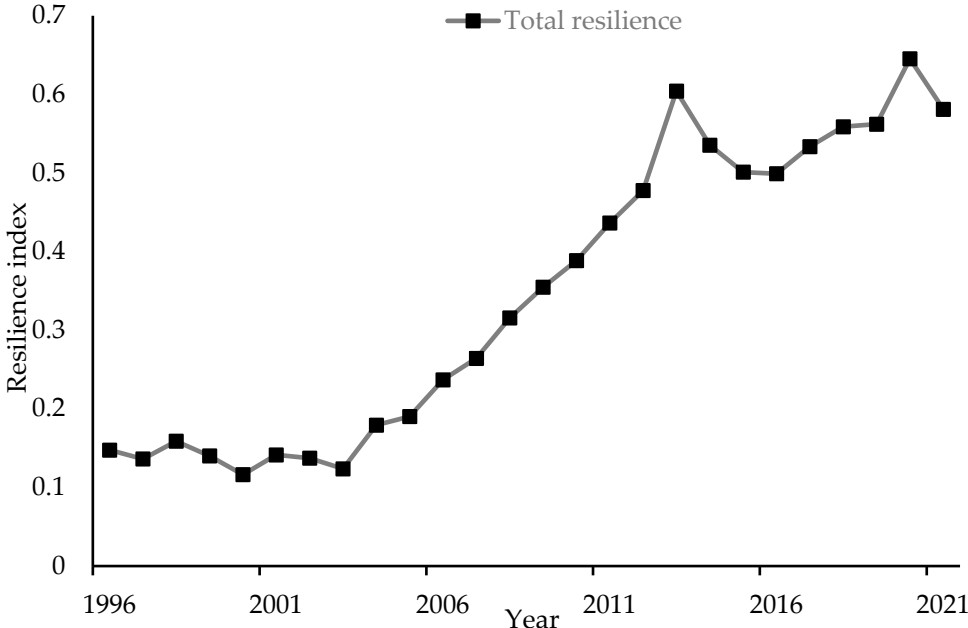

**Figure 2.** Growth of the grain Supply Chain resilience indicator, 1996–2021.

From 1996 to 2021, China's grain supply chain resilience indicator showed a phased fluctuation growth trend. According to the fluctuation of the resilience indicator, the growth trend of the resilience of the grain supply chain can be divided into three phases. 1996–2003 is a stable phase with a low level of resilience of the grain supply chain, which indicates that the comprehensive resilience of China's grain supply chain is at a relatively low level.

From 2004 to 2013, China's grain supply chain was in a growth phase, with successive increases in the level of grain supply chain resilience. From 2014 to 2021, China's grain supply chain was in a high-level growth phase, which indicates that China's grain supply chain resilience has continued to develop in an obvious way during this period.

As shown in Figure 3, there is an overall increase in grain supply chain fracture resilience, supply chain impact resilience, and supply chain synergy resilience from 1996–2021.

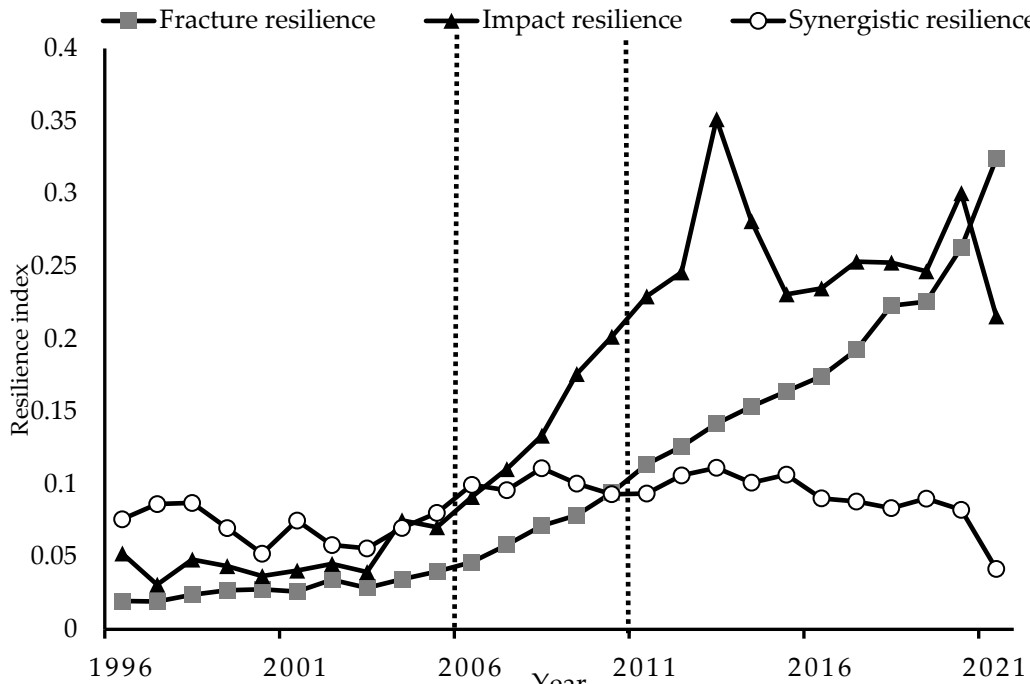

**Figure 3.** Trends in supply chain fracture resilience, supply chain impact resilience, and supply chain synergy resilience, 1996–2021.

As can be seen in Figure 3, supply chain fracture resilience grows steadily, supply chain impact resilience fluctuates more significantly, and supply chain synergy resilience changes more gently. During the study period, with 2006 as the demarcation time point, the overall supply chain synergy resilience is greater than the supply chain impact resilience before 2006, and the supply chain impact resilience is greater than the supply chain synergy resilience after 2006. This indicates that when the grain supply chain faced a major natural disaster in 2003, the role of supply chain synergy resilience is greater than that of the supply chain's resistance to the impact resilience of the ability to resist external shocks. In addition, with 2011 as the cut-off time point, supply chain synergy resilience is greater than supply chain fracture resilience before 2011, and supply chain fracture resilience is greater than supply chain synergy resilience after 2011.

### 5.2. Stationarity Test

One of the basic requirements for establishing a model in econometrics is that the time series must be stationary. Non-stationary time series often lead to spurious regression phenomena, greatly reducing the significance of the regression coefficients and thus affecting the effectiveness of the model. Using SPSS, the following data was tested for stationarity using the ADF test, and all variables were found to be stationary at the 5% confidence level. Differential operation will eliminate trend changes in time series data, resulting in information loss and making the data more difficult to interpret and analyze. Time series data itself is already stable. In order to eliminate heteroskedasticity in time series and linearize its trend, only some variables are transformed into natural logarithms. The test results are shown in Table 3.

**Table 3.** Stabilization of the ADF test.

| Variable | $t$ | $p$ | 1% Critical Value | 5% Critical Value | 10% Critical Value | Conclusion |
|---|---|---|---|---|---|---|
| res | −4.329 | 0.3447 | 0.34 | 0.65 | 0.12 | stationary |
| lnexp | −6.492 | 1.3806 | 1.33 | 1.86 | 0.88 | stationary |
| lnoil | −4.647 | 3.8611 | 3.98 | 4.60 | 2.67 | stationary |
| ex | −3.928 | 0.1383 | 0.14 | 0.16 | 0.12 | stationary |
| urban | −6.116 | 0.4756 | 0.48 | 0.65 | 0.30 | stationary |
| ind | −3.705 | 5.6329 | 4.51 | 11.04 | 3.10 | stationary |
| lninv | −3.103 | 7.7114 | 7.92 | 10.02 | 4.69 | stationary |
| lncap | −3.986 | 8.8180 | 8.85 | 9.20 | 8.43 | stationary |
| lngas | −4.331 | 3.2787 | 3.24 | 4.27 | 2.64 | stationary |

*5.3. Results and Trend Analysis of China's Grain Supply Chain Resilience Measurements*

Table 4 reports the estimated results of the basic model of the impact of external shocks on the resilience of China's grain supply chain. Column (1) does not include control variables, indicating the impact of world grain exports, international oil prices, and the RMB exchange rate on China's grain supply chain resilience. According to the results, the impact of world grain exports on China's grain supply chain is significant at the 1% level, and the coefficient is 0.3710, indicating that for every 1% increase in world grain exports, the total resilience index of the grain supply chain increases by 0.3710%. This suggests that, generally, an increase in the level of world grain exports will enhance the total resilience level of China's grain supply chain. The impact of international oil prices on China's grain supply chain is significant at the 1% level, and the coefficient is −0.2092, indicating that for every 1% increase in international oil prices, the total resilience index of the grain supply chain decreases by −0.2092%. This suggests that, generally, an increase in the level of international oil prices will weaken the total resilience level of China's grain supply chain. The impact of the RMB exchange rate on China's grain supply chain is significant at the 1% level, and the coefficient is 5.2420, indicating that for every 1% increase in international oil prices, the total resilience index of the grain supply chain increases by 5.2420%, indicating that generally, RMB appreciation will increase the total resilience level of China's grain supply chain.

**Table 4.** Impact of external shocks on the resilience of grain supply chains: benchmark regression.

| Variable | (1) res | (2) res | (3) res | (4) res | (5) res | (6) res |
|---|---|---|---|---|---|---|
| lnexp | 0.3710 *** | 0.2818 *** | 0.1954 *** | 0.1662 *** | 0.2391 *** | 0.2347 *** |
|  | (0.0474) | (0.0803) | (0.0341) | (0.0311) | (0.0344) | (0.0304) |
| lnoil | −0.2092 *** | −0.3602 *** | −0.3384 *** | −0.4102 *** | −0.4083 *** | −0.3963 *** |
|  | (0.0218) | (0.0312) | (0.0341) | (0.0350) | (0.0373) | (0.0397) |
| ex | 5.2420 *** | 5.4287 *** | 4.0976 *** | 2.4808 ** | 2.5110 ** | 2.6351 ** |
|  | (1.1068) | (1.1530) | (1.0499) | (1.0014) | (1.2304) | (1.2353) |
| urban |  | 0.5999 | −0.1422 | −0.3202 | −0.3112 | 0.1649 |
|  |  | (0.8758) | (0.7677) | (0.7921) | (0.9478) | (0.8562) |
| ind |  |  | 0.0302 *** | 0.0413 ** | 0.0412 ** | 0.0267 |
|  |  |  | (0.0095) | (0.0151) | (0.0164) | (0.0156) |
| lninv |  |  |  | 0.0425 | 0.0424 | 0.0229 |
|  |  |  |  | (0.0447) | (0.0463) | (0.0415) |
| lncap |  |  |  |  | −0.0065 | 0.0028 |
|  |  |  |  |  | (0.3526) | (0.3105) |
| lngas |  |  |  |  |  | −0.0788 ** |
|  |  |  |  |  |  | (0.0316) |
| _cons | −0.8118 *** | −0.8035 *** | −0.7247 *** | −0.5668 *** | −0.5193 | −0.3336 |
|  | (0.0794) | (0.0813) | (0.0722) | (0.1813) | (2.5915) | (2.2831) |
| $N$ | 26 | 26 | 26 | 26 | 26 | 26 |
| adj. $R^2$ | 0.953 | 0.952 | 0.967 | 0.966 | 0.965 | 0.973 |
| F-statistics | 171.08 | 125.33 | 145.79 | 121.05 | 98.30 | 111.71 |

Note: ***, **, and * indicate significance levels of 1%, 5%, and 10%, respectively; numbers in parentheses are robust standard errors.

Columns (2) to (6) add control variables such as urbanization rate, reasonable industrial structure, rural fixed asset investment, total reservoir capacity, and natural gas prices on the basis of column (1). The core explanatory variables are still significant, indicating that world grain exports and the RMB exchange rate still have a significant positive effect on China's grain supply chain resilience, while international oil prices have a significant negative effect on China's grain supply chain resilience.

### 5.4. Robustness Tests

Grain supply chain resilience is closely related to agricultural policies. The empirical sample of this paper is selected from 1996 to 2021, and in 2006, the country abolished the agricultural tax. In order to illustrate whether the abolition of agricultural tax changes the impact on the resilience of the grain supply chain, therefore, this paper takes the agricultural policy (*poil*) as a dummy variable, which takes the value of 0 before 2006 and the value of 1 in 2006 and later. The coefficients of the world's grain export volume and the exchange rate of the RMB are still significantly positive, and the coefficient of the international petroleum price is significantly negative. This indicates that an increase in the level of the world's grain export volume and an appreciation of the RMB will improve the conclusion that higher levels of world grain exports and RMB appreciation will increase the total resilience of China's grain supply chain, and that higher international oil prices will weaken the total resilience of China's grain supply chain is robust. The specific results are shown in Table 5.

**Table 5.** Consideration of policy influences.

| Variable | (1) res | (2) res | (3) res | (4) res | (5) res | (6) res |
|---|---|---|---|---|---|---|
| lnexp | 0.3579 *** | 0.3539 *** | 0.3744 *** | 0.3664 *** | 0.3528 *** | 0.3713 *** |
| | (0.0535) | (0.0883) | (0.0415) | (0.0182) | (0.0353) | (0.0312) |
| lnoil | −0.2893 *** | −0.4593 *** | −0.2574 *** | 0.3431 *** | 0.3414 *** | 0.1005 ** |
| | (0.0264) | (0.0356) | (0.0378) | (0.0393) | (0.0418) | (0.0463) |
| ex | 5.0250 *** | 5.2027 *** | 3.9424 *** | 2.4614 ** | 2.4949 ** | 2.6011 ** |
| | (1.1895) | (1.2283) | (1.1067) | (1.0471) | (1.0942) | (1.2135) |
| urban | | 0.6426 | −0.1026 | −0.2784 | −0.2685 | 0.1564 |
| | | (0.8920) | (0.7850) | (0.8162) | (0.9766) | (0.8826) |
| ind | | | 0.0299 *** | 0.0404 ** | 0.0403 ** | 0.0268 |
| | | | (0.0097) | (0.0156) | (0.0170) | (0.0161) |
| lninv | | | | 0.0398 | 0.0397 | 0.0237 |
| | | | | (0.0462) | (0.0478) | (0.0429) |
| lncap | | | | | −0.0072 | 0.0033 |
| | | | | | (0.3611) | (0.3197) |
| lngas | | | | | | −0.0806 ** |
| | | | | | | (0.0338) |
| poil | 0.0238 | 0.0262 | 0.0196 | 0.0152 | 0.0152 | −0.0067 |
| | (0.0425) | (0.0432) | (0.0362) | (0.0368) | (0.0379) | (0.0348) |
| _cons | −0.7476 *** | −0.7321 *** | −0.6721 *** | −0.5359 ** | −0.4831 | −0.3455 |
| | (0.1403) | (0.1435) | (0.1217) | (0.1999) | (2.6556) | (2.3514) |
| N | 26 | 26 | 26 | 26 | 26 | 26 |
| adj. $R^2$ | 0.952 | 0.951 | 0.965 | 0.965 | 0.963 | 0.971 |
| F-statistics | 124.38 | 97.33 | 117.25 | 99.25 | 82.02 | 93.68 |

Note: ***, **, and * indicate significance levels of 1%, 5%, and 10%, respectively; numbers in parentheses are robust standard errors.

In this section, the robustness of the impact of external shocks on the level of aggregate resilience of the grain supply chain is analyzed by testing the regression of external shocks on the resilience of the grain supply chain shocks (*ri*), aggregate grain production (*grain*), and per capita grain production (*grainp*). The results of the test are shown in Table 6, which shows that world grain exports and the RMB exchange rate have a significant positive effect on the resilience of grain supply chain shocks, total grain production, and per capita grain production, while international oil prices have a significant negative effect on the resilience of grain supply chain shocks, total grain production, and per capita grain production. This result is generally consistent with Table 4, indicating that the results in Table 4 are robust.

**Table 6.** Replacement of explanatory variables.

| Variable | (1)<br>ri | (2)<br>ri | (3)<br>lngrain | (4)<br>lngrain | (5)<br>lngrainp | (6)<br>lngrainp |
|---|---|---|---|---|---|---|
| lnexp | 0.1191 *** | 0.1121 *** | 0.4679 *** | 0.1721 *** | 0.3320 *** | 0.2112 *** |
|  | (0.0365) | (0.0297) | (0.0719) | (0.0388) | (0.0739) | (0.0386) |
| lnoil | −0.0947 *** | −0.0629 *** | −0.0844 ** | −0.0916 * | −0.0992 *** | −0.0924 * |
|  | (0.0168) | (0.0188) | (0.0331) | (0.0508) | (0.0340) | (0.0506) |
| ex | 4.0325 *** | 3.5032 ** | 7.3662 *** | 5.2575 *** | 7.6020 *** | 5.0575 ** |
|  | (0.8522) | (1.3840) | (1.6797) | (1.1182) | (1.7267) | (2.1049) |
| urban |  | 0.1398 |  | −1.3004 |  | −1.5342 |
|  |  | (0.8381) |  | (1.0963) |  | (1.0916) |
| ind |  | 0.0081 |  | 0.0840 *** |  | 0.0856 *** |
|  |  | (0.0153) |  | (0.0200) |  | (0.0199) |
| lninv |  | 0.0323 |  | 0.1538 ** |  | 0.1452 ** |
|  |  | (0.0406) |  | (0.0531) |  | (0.0529) |
| lncap |  | 0.0030 |  | −0.2761 |  | −0.2759 |
|  |  | (0.3040) |  | (0.3976) |  | (0.3959) |
| lngas |  | −0.0616 * |  | −0.0482 |  | −0.0518 |
|  |  | (0.0310) |  | (0.0405) |  | (0.0403) |
| _cons | −0.5617 *** | −0.1885 | 9.6536 *** | 12.6021 *** | 4.9762 *** | 7.9147 ** |
|  | (0.0611) | (2.2350) | (0.1204) | (2.9233) | (0.1238) | (2.9108) |
| N | 26 | 26 | 26 | 26 | 26 | 26 |
| adj. $R^2$ | 0.907 | 0.911 | 0.928 | 0.970 | 0.889 | 0.957 |
| F-statistics | 81.87 | 33.09 | 108.93 | 102.10 | 67.99 | 69.84 |

Note: ***, **, and * indicate significance levels of 1%, 5%, and 10%, respectively; numbers in parentheses are robust standard errors.

Considering that the resilience of the grain supply chain is a gradual process, external shocks have a certain lag effect on the grain supply chain. Therefore, the world grain export volume, international oil price, and RMB exchange rate are lagged by one period and re-estimated using the least squares method, as shown in Table 7. After considering the lag effect of explanatory variables, the coefficients of world grain export volume and RMB exchange rate are still significantly positive, and the coefficient of international oil price is significantly negative. This indicates that the increase in the level of world grain export volume and the appreciation of RMB will enhance the overall resilience level of China's grain supply chain. The conclusion that the increase in international oil price will weaken the overall resilience level of China's grain supply chain is robust.

**Table 7.** Results of the robustness test considering the lag effect.

| Variable | (1)<br>res | (2)<br>res | (3)<br>res | (4)<br>res | (5)<br>res | (6)<br>res | (7)<br>res |
|---|---|---|---|---|---|---|---|
| lnexp(−1) | 0.3870 *** | 0.5184 *** | 0.3286 *** | 0.4584 *** | 0.6070 *** | 0.9122 *** | 0.7577 *** |
|  | (0.0524) | (0.0309) | (0.0584) | (0.0855) | (0.0962) | (0.0990) | (0.0309) |
| lnoil(−1) | −0.2796 *** | −0.2654 *** | −0.2510 *** | −0.2494 *** | −0.2492 *** | −0.2548 *** | −0.2455 *** |
|  | (0.0233) | (0.0344) | (0.0331) | (0.0336) | (0.0345) | (0.0351) | (0.0427) |
| ex(−1) | 3.8494 *** | 4.2726 *** | 3.1443 *** | 2.9396 *** | 2.6410 ** | 2.8052 * | 2.2090 * |
|  | (1.1848) | (1.2402) | (1.0975) | (1.0643) | (1.2979) | (1.4725) | (1.3261) |
| urban |  | 1.0658 | 0.4173 | 0.1390 | −0.0099 | 0.3280 | 0.3845 |
|  |  | (0.9691) | (0.8358) | (0.9378) | (1.0601) | (1.1233) | (1.1620) |
| ind |  |  | 0.0297 *** | 0.0362 ** | 0.0365 ** | 0.0285 | 0.0272 |
|  |  |  | (0.0095) | (0.0135) | (0.0139) | (0.0164) | (0.0171) |
| lninv |  |  |  | 0.0322 | 0.0305 | 0.0288 | 0.0217 |
|  |  |  |  | (0.0465) | (0.0480) | (0.0482) | (0.0525) |
| lncap |  |  |  |  | 0.1065 | 0.1071 | 0.0876 |
|  |  |  |  |  | (0.3185) | (0.3197) | (0.3318) |
| lngas |  |  |  |  |  | −0.0318 | −0.0298 |
|  |  |  |  |  |  | (0.0339) | (0.0352) |

**Table 7.** *Cont.*

| Variable | (1) res | (2) res | (3) res | (4) res | (5) res | (6) res | (7) res |
|---|---|---|---|---|---|---|---|
| poil | | | | | | | 0.0185 |
| | | | | | | | (0.0454) |
| _cons | −0.8114 *** | −0.8202 *** | −0.6910 *** | −0.5674 *** | −1.3621 | −1.2320 | −1.0828 |
| | (0.0848) | (0.0848) | (0.0821) | (0.1968) | (2.3858) | (2.3984) | (2.4905) |
| N | 25 | 25 | 25 | 25 | 25 | 25 | 25 |
| adj. $R^2$ | 0.946 | 0.947 | 0.963 | 0.962 | 0.960 | 0.960 | 0.957 |
| F-statistics | 142.19 | 108.01 | 125.81 | 102.05 | 83.17 | 72.37 | 60.99 |

Note: ***, **, and * indicate significance levels of 1%, 5%, and 10%, respectively; numbers in parentheses are robust standard errors.

## 6. Discussion and Conclusions

This paper draws on the definition of resilience in physics to construct a comprehensive evaluation indicator system for the resilience of the grain supply chain. The system includes supply chain fracture resilience, impact resilience, and synergy resilience as secondary indicators. It adopts the entropy value method to calculate China's grain supply chain total resilience indicator, fracture resilience indicator, impact resilience indicator, and synergy resilience indicator for each year from 1996 to 2021. Additionally, it constructs an econometric model to investigate the effects of external shocks, such as supply shock, cost change shock, and exchange rate fluctuation shock, on China's grain supply chain resilience. The study also constructs an econometric model to investigate the impact of external shocks, such as external supply shocks, cost shocks, and exchange rate fluctuation shocks, on the resilience of China's grain supply chain. The study found that: from the view of China's grain supply chain resilience indicator growth trend map, China's total grain supply chain resilience in 1996–2021 was a stage-like fluctuation growth trend, mainly divided into three growth stages, respectively, a low level of stability stage, continuous growth stage, and a high level of stability stage; from the view of the resilience trend change map, the supply chain fracture resilience steadily increased in 1996–2021, the supply chain impact resilience level fluctuated significantly, the supply chain resilience level fluctuated markedly, and the supply chain impact resilience level fluctuated markedly. Supply chain impact resilience level fluctuates significantly, and supply chain synergy resilience changes are flat.

The main reason for the high level of growth of China's grain supply chain in 2014–2021 is reflected in the fact that, on the one hand, the state implemented reforms from both a policy and system perspective to prevent the impact of external shocks on China's grain supply chain. This was done in order to reduce the uncertainty caused by external shocks and to restore the level of grain supply chain resilience in a timely manner. In 2015, China established the grain risk fund system to promote the stable growth of grain production and maintain the normal grain circulation order. In 2016, China further deepened the reform of the grain supply chain system, improving production and marketing as well as purchase and marketing reform. Therefore, at this stage, China's grain supply chain resilience is in the recovery stage. On the other hand, pay attention to the scientific and technological development of the grain supply chain and cultivate excellent talents in the grain industry. At this stage, the research and development capacity of China's grain supply chain system has been improving year by year, and the number of effective invention patents has increased from 4858 in 2014 to 13,432 in 2021. The growth rate of senior agricultural technicians during this period was 6.06%, and scientific and technological innovation serves as the driving force for the adjustment and renewal of the grain supply chain and its long-term development. This not only promotes the recovery of the resilience level of China's grain supply but also enables further improvement of the resilience level.

In exploring the external shocks to the resilience of China's grain supply chain, it is found that world grain exports and the RMB exchange rate have a significant positive

impact on the total resilience level of China's grain supply chain, and international oil prices have a significant negative impact on the total resilience level of China's grain supply chain.

Based on these findings, this paper draws the following policy implications.

First, to ensure stable production and supply in the grain market. We will start by improving the overall quality of China's grain, so as to ensure the effective supply of Chinese grain and reduce China's dependence on grain exports. The first is to hold fast to the bottom line of arable land, resolutely sticking to the bottom line of 1.8 billion mu of arable land, and accelerating agricultural capital construction in an effort to raise the level of land productivity. The second is to promote the development of the seed industry by building a new variety selection and breeding system that combines production, breeding, and promotion. This includes the basic theory of seed sources and focuses on breakthroughs in the yields of soybeans and corn. The goal is to consolidate China's grain security and lay a solid theoretical and practical foundation. Third, the strength of agricultural support needs to be increased. The policy should be implemented in place and must consider the minimum purchase price of grain crops and related policies. It should also take into account the international rules of constraints and constantly optimize the subsidy policy for corn and soybean output. This should be done practically, from the interests of the farmers, to ensure that they are incentivized to grow grain. This will provide effective protection against the risks of the international grain market, while maintaining stable grain production.

Secondly, the structure of foreign trade in grain should be improved. China's four major grain crops have almost all been net imports in recent years, and in order to optimize the structure of China's foreign trade in grains, it is necessary to implement a diversified import strategy. In particular, since the current domestic soybean production is small and mainly imported, the only way to optimize the structure of China's foreign trade in grain is to adopt a diversified import strategy and to share as much as possible the adverse effects of the changes in the economic situation of the world's countries on the domestic grain market. In addition, the use of mergers and acquisitions and expansion around the world can extend their business areas to all areas of the agricultural business and gradually establish a global grain industry chain. This is helpful in better grasping the world grain market and realizing the improvement of the resilience of the grain supply chain.

Third, actively responding to international commodity price shocks. As can be seen from the above, international oil prices have a significant negative impact on the level of total resilience of China's grain supply chain. Therefore, coping with the impact of global price increases is an urgent issue in China's current economic development. The following measures can be taken: first, pay close attention to the price changes of various major products and formulate a long-term development strategy to maintain sufficient stocks of various major products, especially energy products. Secondly, foreign futures markets should be monitored, and further improvements should be made to domestic futures markets. Most international primary products are priced through a futures market that can forecast spot prices and guide their movements. By monitoring the dynamics of the international futures market and conducting further research on the domestic and foreign futures markets, it can help us better understand and grasp the international price trends. Third, the linkage between agricultural futures and spot should be strengthened. China's grain production has a large time span and obvious seasonal characteristics, and its response to price changes lags behind, making it difficult to effectively respond to market risks. However, its price discovery and hedging role can just make up for this shortcoming. In China's grain and agricultural enterprises, the use of futures for hedging can effectively avoid and eliminate all kinds of hedging risks and safeguard their production benefits. To this end, the linkage between domestic grain prices and futures market prices should be further strengthened, giving full play to the forecasting role of futures prices on grain prices, and realizing the smooth operation of grain prices in order to reduce the volatility of grain prices.

**Author Contributions:** Conceptualization, T.Z. and G.Z.; methodology, T.Z.; software, G.Z.; validation, T.Z., G.Z. and S.C.; formal analysis, T.Z.; investigation, T.Z.; resources, T.Z.; data curation, G.Z.; writing—original draft preparation, S.C.; writing—review and editing, G.Z.; visualization, G.Z.; supervision, S.C.; project administration, T.Z.; funding acquisition, T.Z. All authors have read and agreed to the published version of the manuscript.

**Funding:** 2022 Hebei Provincial Social Science Fund "Research on the Coupling Mechanism and Driving Factors of Hebei Digital Village and the Whole Agricultural Industry chain" (HB22YJ050).

**Institutional Review Board Statement:** Not applicable.

**Informed Consent Statement:** Not applicable.

**Data Availability Statement:** The data presented in this study are available on request from the corresponding author.

**Conflicts of Interest:** The authors declare no conflict of interest.

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
