# Peer review of "A Study on the Impact of External Shocks on the Resilience of China’s Grain Supply Chain"

_sustainability, doi:10.3390/su16030956_

Round 1

Reviewer 1 Report

Comments and Suggestions for Authors

Resilience to fractures in the cereal supply chain is a current, central and very important issue for any country, whether developed or developing. This research paper looks at this issue applied to the reality of China, a country that is currently at the centre of global development.

Issues such as resilience, fluctuation, the impact of external shocks and the influence of oil prices are realities that should concern leaders and public managers, otherwise there will be strong social and economic impacts on the day-to-day life of any nation. After analysing the above-mentioned issues, the authors were able to come up with a series of suggestions for guaranteeing cereal production and supply, without forgetting foreign trade issues and identifying how to mitigate international shocks to cereal prices.

As well as being up-to-date, this work deserves the attention of leaders and managers because it sheds light on a strategic issue for any nation. I recognise the academic relevance of the study and therefore consider it, as well as being relevant, a must-read exercise. From a structural point of view, it is a systematic work with organised and very objective ideas for explaining the problem in question. The time frame in question is very relevant, which allows me to state some ideas with academic substance and propriety.

Comments on the Quality of English Language

 Minor editing of English language required.

Reviewer 2 Report

Comments and Suggestions for Authors

1.       The interpretation of some tertiary indicators deserves clarification. Is it true that average labor productivity (grain production / average number of people in industry) is a good measure of production capacity? The same applies to warehouse capacity: “grain production + net imports – grain sales” is a measure of flow, not the static capacity of warehouses.

2.       The interpretation of the disaster resilience indicator is unclear. The effects of disasters are mainly local or regional and of different natures (droughts, floods, earthquakes...). Was there any type of aggregation at the national level?

3.       It would be useful to understanding Table 1 to include the measurement units for each of the variables.

4.       Why were hypotheses 1-3 tested using different regressions, rather than a single multiple regression? It is possible that the three variables exp, oil and ex are not linearly independent, so that the regressions fail to capture possible cross effects of these variables.

5.       All variables are time series. Was the stationarity of the series tested, to allow the use of regressions that disregard lagged variables? Some high adjusted R2 suggest the presence of serial correlations.

Reviewer 3 Report

Comments and Suggestions for Authors

The paper is well structured, but lacks important details that would help readers understand exactly what has been done. There are also some concerns with the dataset used to conduct the analysis.

Major Comments

1- The concept of resiliance is central to the paper and should therefore be defined much earlier on. We do not need to know that there are multiple definitions across multiple fields, but instead want to know how you define it in this paper. This would help better understand what you are doing and give context to the borad literature review that follows.

2- The literature review reads more like a general survey as opposed to a review of the relevant literature for this paper. It would be potentially useful to state your methodology first so that in provides context to the discussion. 

3- More information about the data that you use to estimate the models is required. You state that they are mainly from the China Statistical Yearbook, China Agricultural Statistical Yearbook. But exactly which series are used? If the series are available at high frequencies, such as the price of oil, then how do you convert them to an annual series? Such infromation can be provided in a Table. 

4- More detail on how your resilience indicator is constructed is essential. Do you use principle components? A weighted average? A simple average? Such details are essential in determining the credibility of the index. 

5- The paper uses annual observations from 1996-2021. This means that there are only 26 observations per series used. It is not clear to me which estimator has been used to estimate the model, but if it is maximum likelihood, then the small sample size raises concerned about the asymptotics of the estimator.

6-In economics, a 'shock' is typically defined as an unexpected event. Shocks can be observed, e.g., an earthquake, or unobserved, e.g., an unanticipated increase in the price of oil. In the latter case, an econometric model is required to identify the shock by decomposing the observed time series into expected and unexpected components. This is important. The variables used in the regression are observable time series and do not match your hypothesis which relates to shocks. You therefore need to create economic shock series and then use these in your analysis. The following chapter which clarifies this in detail: Ramey, V. A. (2016). Macroeconomic shocks and their propagation. Handbook of macroeconomics, 2, 71-162.

7- Related to the above point. The dynamic analysis of shocks in time series regressions can be done using local projections framework. For more context, see: Jordà, Ò. (2005). Estimation and inference of impulse responses by local projections. American economic review, 95(1), 161-182. 

Minor Comments

1- The paper is motivated by noting that grain production is important for national security. Why? Many countries do not produce grain. Are these countries at risk? If China's government declares grain production as important for national security then you should reference such policies to convince readers of its national importance. 

2- The term 'Crown Pneumonia epidemic' is not widely used in the Western world. I therefore think it's important to add conext. A simple footnote that the Chinese translation of COVID-19 is 新冠肺炎 (xinguan feiyan), meaning “new crown pneumonia”, would be useful here.

3- The Russian-Ukrainian 'situation' is an official war, so should be declared as such.

4- The term "European breadbasket" is used without explanation. It would be useful to note that this related to Ukraine. 

5- There has only been one Great Recession - which was in 2008-09 (not 4). 

Comments on the Quality of English Language

The English is OK. 

Round 2

Reviewer 2 Report

Comments and Suggestions for Authors

I will insist on one specific point: the parameters of equation (1) were not estimated. None of the models presented in Table 3 correspond to equation (1). Furthermore, to have confidence in the estimates, the stationarity of the time series must be guaranteed. See, for example, the classic C.W. Granger, P. Newbold. Spurious regressions in econometrics. J. Econ., 2 (2) (1974), pp. 111-120, 10.1016/0304-4076(74)90034-7

Reviewer 3 Report

Comments and Suggestions for Authors

I'm glad the comments were useful and improved the paper. I still think that a few points should be addressed:

1. Literature Review: I agree that this section is fine for a dissertation, but this is an academic publication. I therefore suggest that you rewrite the section accordingly. 

2. Estimator properties: I accept that data availability is an issue, however, the questions of whether the estimators are reliable with such a short sample (26 observations) has not been addressed. Many of the series in the model are available at higher frequencies. One method to gain more data points is to thus use a mixed frequency regression. Another is to disaggregate the annual series to a higher frequency, such as the quarterly frequency. Of course, this has its own issues, but at least then the estimator will be reliable...

3. Dynamics: I'm still not convinced that the main regression accounts for dynamics. Why are there no lags in the model?

4. Identification: It is still not clear to me that the parameters in the model are well identified given the potential bias in the estimator due to simultaneity issues. 

Round 3

Reviewer 2 Report

Comments and Suggestions for Authors

The article has been substantially improved in this version. It would be worth making two small clarifications: (i) confirm that the ADF test refers to level variables, not differences; (ii) point to the limitation arising from the small sample size (n=24).

Reviewer 3 Report

Comments and Suggestions for Authors

None
